# Linking Periodontitis with Inflammatory Bowel Disease through the Oral–Gut Axis: The Potential Role of *Porphyromonas gingivalis*

**DOI:** 10.3390/biomedicines12030685

**Published:** 2024-03-19

**Authors:** Xinyi Huang, Yilin Li, Jun Zhang, Qiang Feng

**Affiliations:** Shandong Key Laboratory of Oral Tissue Regeneration, Shandong Engineering Laboratory for Dental Materials and Oral Tissue Regeneration, Department of Orthodontics, School and Hospital of Stomatology, Cheeloo College of Medicine, Shandong University, No. 44-1 Wenhua Road West, Jinan 250012, China; 201900242007@mail.sdu.edu.cn (X.H.); leeyiling@126.com (Y.L.)

**Keywords:** *Porphyromonas gingivalis*, inflammatory bowel disease, oral–gut axis, periodontitis

## Abstract

Periodontitis and inflammatory bowel disease (IBD) are both chronic inflammatory diseases that are characterized by abnormal host immune responses and microbiota dysbiosis. Emerging evidence implies potential associations between periodontitis and IBD. *Porphyromonas gingivalis* (*P. gingivalis*), a primary cause of periodontitis, is thought to play a role in the development of IBD through the oral–gut disease axis. However, the precise mechanisms of its involvement remain enigmatic. In this narrative review, we begin with a discussion of the bidirectional relationship between periodontitis and IBD and the involvement of *P. gingivalis* in each of the two diseases. Further, we summarize the possible routes by which *P. gingivalis* links periodontitis and IBD through the oral–gut axis, as well as the underlying mechanisms of its involvement in the pathogenesis of IBD. Collectively, *P. gingivalis* participates in the progression of IBD through gut dysbiosis, impairment of the intestinal barrier, release of inflammatory mediators, and disturbance of the immune response. The above findings may provide new insights for exploring novel biomarkers and potential therapeutic approaches for IBD.

## 1. Introduction

Periodontitis represents the inflammatory response of the oral periodontal tissues, including the gingiva, alveolar bone, and periodontal membrane [1]. Swelling and redness of the gingiva are usually the earliest symptoms of periodontitis, which may be followed by attachment loss and alveolar bone resorption. Severe periodontitis can lead to tooth loosening or loss, negatively affecting general health [2,3].

Notably, individuals with periodontitis often suffer from other complications, one of which is inflammatory bowel disease (IBD) [4,5]. IBD is a chronic inflammatory response of the gastrointestinal tract. Clinically, it primarily manifests in two forms: Crohn’s disease (CD) and ulcerative colitis (UC) [6]. CD is a gastrointestinal tract lesion that often presents with parenteral symptoms in the skin, joints, or eyes (common examples include erythema nodosum, pauciarticular large joint arthritis, axial arthropathies, etc.) and is also accompanied by immune dysfunction [7,8]. UC, on the other hand, is an idiopathic, chronic condition affecting the colonic mucosa [9]. To date, the pathogenesis of IBD remains unclear. However, studies have identified various risk factors that contribute to its progression, with the disturbance of the gut microbiota emerging as a potentially important influence [6].

Accumulating epidemiological evidence has shown a significant association between periodontitis and IBD. As mentioned earlier, the microbiota contributes to both periodontitis and IBD. Could there be a connection between the oral and gut microbiota that forms a bridge between periodontitis and IBD? The recent concept of the “oral–gut axis” may hold the answer, suggesting the involvement of oral pathobionts in the pathogenesis of IBD [5,10]. 

*Porphyromonas gingivalis* (*P. gingivalis*) is widely regarded as a key pathogen in the development of periodontitis [11]. With the advancements in microbiology, oral medicine, and related research techniques, the involvement of *P. gingivalis* in systemic diseases has been discovered [12]. Multiple studies have revealed that *P. gingivalis* is a risk factor for numerous systemic diseases, such as cardiovascular diseases [13], diabetes [14], and Alzheimer’s disease [15]. Emerging evidence suggests that *P. gingivalis* is involved in the pathogenesis of IBD through the oral–gut axis [16,17], but the precise mechanisms remain unclear. In this article, we reviewed recent studies examining the intricate interactions between periodontitis and IBD, as well as the evidence for the engagement of *P. gingivalis*, a keystone pathogen of periodontitis, in the progression of IBD. Further, we summarized the possible routes through which *P. gingivalis* links the two diseases through the oral–gut axis and the potential mechanisms for its involvement in the pathogenesis of IBD.

## 2. Role of *P. gingivalis* as a Keystone Pathogen of Periodontitis

*P. gingivalis* is a Gram-negative, obligate anaerobic bacteria. It has been detected in 85.75% of subgingival plaque in patients with chronic periodontitis [18]. Researchers have successfully induced periodontitis in rhesus monkeys by implanting *P. gingivalis* into their subgingival microbiota [19]. As one of the major pathogens in periodontitis, *P. gingivalis* exhibits a strong correlation between its prevalence and the severity of periodontal disease. Even in low abundance, *P. gingivalis* has the ability to induce chronic periodontitis by remodeling the commensal bacterial community, resulting in dysbiosis, thus earning its status as a keystone pathogen of periodontitis [20]. Once colonized, *P. gingivalis* disrupts the host immune defense and promotes inflammation, leading to alterations in the abundance and composition of the subgingival microbiota and ultimately dysbiosis [21].

*P. gingivalis* is capable of expressing and releasing several virulence factors, including lipopolysaccharides (LPS), Trichoderma, gingipains, tetratricopeptide repeat sequence protein, extracellular polysaccharides, and the hemoglobin uptake system [22,23]. These virulence factors are capable of mediating immune dysfunction in periodontal tissues. This leads to immune cell infiltration and inflammation, ultimately destroying periodontal tissue [23]. The destructed tissues release nutrients, including degraded collagen, sources of amino acids and iron, and heme-containing compounds. These nutrients penetrate the gingival crevice through gingival crevicular fluid (GCF), promoting the growth of periodontitis-associated microorganisms. This, in turn, further exacerbates the dysbiosis associated with periodontitis [24].

In addition, *P. gingivalis* belongs to the category of intracellular pathogenic bacteria. By invading multiple human cells, *P. gingivalis* adjusts its expression pattern to evade immune surveillance, allowing it to serve as a reservoir for persistent infection and inflammation induction [25,26]. Moreover, *P. gingivalis* demonstrates the capacity to interact with host cells, resulting in its long-term colonization within the oral cavity [25]. 

## 3. Bidirectional Relationship between Periodontitis and IBD

### 3.1. Epidemiological Evidence

Three meta-analysis reports integrated observational studies to reveal a potential association between periodontitis and IBD. These studies showed a correlation between periodontitis with CD and UC, with pooled odds ratios ranging from 1.72 to 3.64 for CD and 2.39 to 5.37 for UC [27,28,29]. Furthermore, a study conducted by Schmidt et al. [30] revealed that patients with IBD exhibited more severe periodontitis, characterized by a higher clinical attachment loss (CAL) and gingival bleeding. On the other hand, the prevalence of IBD is elevated in patients with periodontitis. Data from the Women’s Health Initiative observational cohort study revealed an association between poorer oral health and IBD [31]. Another national cohort study conducted in Korea suggested a higher likelihood of UC development among patients with periodontitis, especially among smokers aged ≥65 years. However, the association between periodontitis and CD was not statistically significant in this study [32]. A study from Taiwan showed that patients with periodontitis had a significantly increased likelihood of developing secondary UC [33]. Interestingly, a cohort study including more than 20,000 individuals demonstrated a reduced risk of IBD among patients with periodontitis [34]. This reversed relationship may be influenced by factors such as diet and ethnicity, and it is important to delve deeper into the exact causes. 

It must be noted that the existing cohort studies have not been able to explain the exact interaction between periodontitis and IBD. The primary reason for this incomplete understanding is the predominant reliance on cross-sectional data, lacking the support of longitudinal observations to capture the dynamic nature of the association over time. Although longitudinal observations were provided in some of the larger cohort studies, the precise causality between periodontitis and IBD remains unclear [32,33,34,35]. 

### 3.2. Microbiological Associations between Periodontitis and IBD

The oral cavity and the gut, being the two ends of the digestive tract, are inhabited by unique microbiota that play a pivotal role in human health and disease development [36]. Emerging evidence has demonstrated a strong correlation between oral microbiota and gut diseases. Changes in the gut and oral microbiota of IBD patients provide additional support for the link between periodontitis and IBD [5,37]. Researchers have discovered an abnormal enrichment of typical oral resident bacteria in the intestinal lumen contents and intestinal mucosal tissues of patients suffering from intestinal disease [38]. Therefore, it is conceivable that the translocation of oral pathobionts to the gut and their ectopic colonization may contribute to intestinal disease [39]. Gevers et al. [40] revealed that patients with intestinal inflammation exhibit a significant enrichment of oral bacteria in the gut, including pathogens associated with periodontitis. Similarly, Atarashi et al. [41] demonstrated that periodontal pathogens can ectopically translocate from the oral cavity to the gut, inducing gut dysbiosis and an intestinal immune response that exacerbates intestinal inflammation, suggesting a unique association between periodontitis and IBD. Meanwhile, individuals with IBD have unique oral characteristics compared to healthy individuals. Specifically, they exhibit dysregulation of the oral microbiome, including salivary, dental plaque, tongue, and buccal mucosal microbiota [42,43], and this oral dysbiosis is further associated with the severity of IBD [44]. Moreover, compared to the older population, a higher abundance of the phylum *Bacteroidetes* was found in the oral microbiome of the younger population [45], which is one of the characteristics of the oral microbiome in IBD patients [42,43], and this may account for the higher prevalence of IBD in young people. These findings suggest a potential link between periodontitis and IBD through microbial communication.

While the existing epidemiological evidence remains controversial in explaining the interaction between periodontitis and IBD, biological evidence suggests that patients with IBD exhibit significant oral dysbiosis. Furthermore, oral bacteria, particularly periodontitis-related pathogens, contribute to the development of IBD through gut translocation and ectopic colonization. Therefore, it is necessary to conduct more high-quality, standardized cohort studies to reveal the exact relationship between periodontitis and IBD.

## 4. Potential Involvement of *P. gingivalis* in the Progression of IBD

Several lines of evidence implicate a potential association between *P. gingivalis* and IBD. Stein et al. [46] analyzed periodontal pathogens in 147 subgingival plaque samples from CD patients, finding that 62.6% of them contained *P. gingivalis*. By analyzing the microbial macrogenome, Lee et al. [47] found that the abundance of Porphyromonadaceae in fecal samples from patients with CD was significantly higher compared to control volunteers. In vivo experiments have also provided evidence to support the potential association between *P. gingivalis* and IBD. Researchers observed a significant increase in inflammatory infiltrating cells within the lamina propria of colonic tissue in *P. gingivalis*-treated mice compared to PBS controls [48]. Similarly, in the Dextran sulfate sodium (DSS)-induced colitis model, mice treated with *P. gingivalis* exhibited a more severe clinical presentation, characterized by an increased disease activity index (DAI) score and a shortened colon length. On a histological level, it was observed that the administration of *P. gingivalis* resulted in severe active inflammation with extensive epithelial loss, marked crypt destruction, and dense cellular infiltrations [16,17,47]. These findings demonstrate that *P. gingivalis* may aggravate colitis induced by DSS. Collectively, the evidence supports the potential involvement of *P. gingivalis* in the progression of IBD. 

## 5. Linking PD with IBD: Gut Translocation of *P. gingivalis*

Given the significant abundance of Porphyromonadaceae detected in fecal samples from IBD patients, which suggests a potential role for *P. gingivalis* in the pathogenesis of IBD [47], researchers have analyzed the gut translocation pathways of *P. gingivalis*. Two potential routes have been proposed for the migration of *P. gingivalis* from the oral cavity to the intestine, possibly contributing to the pathogenesis of IBD.

### 5.1. Enteral Dissemination

Each day, individuals swallow approximately 600 times and produce around 1.5 L of saliva. This saliva transports enzymes, effector cytokines, oral microorganisms, and various inflammatory cells toward the intestine. However, these microorganisms, cells, and their respective products rarely reach and colonize in a healthy intestine due to the presence of gastric acidity and intestinal barriers [10,49]. Currently, gastric acidity is considered to be the largest barrier to the translocation of oral bacteria to the gut [50,51]. It is estimated that more than 99.9% of swallowed bacteria of oral origin cannot survive in the stomach owing to its acidic environment [52,53]. Conversely, *P. gingivalis* has the unique ability to tolerate acidic environments and pass through the gastric barrier, offering it a natural advantage for gut translocation [54]. In addition, the number of ingested oral pathobionts must reach a threshold for their successful translocation from the oral cavity to the intestine. Quantitative analysis reveals that patients with severe periodontitis can swallow up to 1012–1013 *P. gingivalis* per day [55,56,57], which provides favorable prerequisites for its gut translocation. The disruption of the colonization resistance of the gut resident microbiota also serves as a favorable condition for the invasion of oral pathobionts [58]. By establishing the human oral microbiota-associated (HOMA) mouse model, Li et al. [59] demonstrated that *P. gingivalis* plays a key role in competing for colonization with resident bacteria in the small intestine. 

The enteral dissemination of *P. gingivalis* is further supported by in vivo experiments. Researchers observed a significant inflammatory infiltration in the gut of mice in the treatment group that were orally administered *P. gingivalis*, compared to those in the control group [48]. Likewise, in the DSS-induced colitis model, mice orally administered *P. gingivalis* exhibited more severe inflammation at both clinical and histologic levels [16,17,47]. Collectively, these findings demonstrate that *P. gingivalis* can induce or exacerbate inflammation in the gut through enteral dissemination.

### 5.2. Hematogenous Dissemination

Studies have revealed that both pathological conditions and routine dental activities, including dental treatment, brushing, flossing, and daily chewing, may lead to mechanical damage to the oral cavity. This damage allows oral bacteria and their toxins to spread into the circulation, causing systematic chronic inflammation [60,61,62,63]. An essential clue is the effect of LPS on neutrophils in the circulation. In the context of periodontitis, LPS can induce changes in the neutrophil phenotype and enhance the inflammatory response in the circulation [64]. However, the detailed mechanism behind the altered neutrophil status and circulation hyperinflammation in periodontitis remains controversial. Whether these changes are due to endotoxemia caused by the entry of LPS into the circulation or due to LPS-related immune training of infiltrating neutrophils in the oral microenvironment needs to be further verified [65].

Furthermore, *P. gingivalis* may achieve hematogenous migration by invading and surviving in dendritic cells (DCs) and macrophages. Specifically, *P. gingivalis* enters macrophages via complement receptor 3 (CR3) in cholesterol-rich lipid rafts, which is mediated by FimA fimbriae [66,67]. To survive in DCs, the Mfa1 fimbriae of *P. gingivalis* interact with the C-type lectin DC-specific ICAM-3 grabbing non-integrin (DC-SIGN), facilitating its invasion into these cells [68,69]. Another pathogenic study similarly characterized the role played by different species of fimbriae in the invasion of *P. gingivalis* into DCs [70]. Mfa1 fimbriae inhibit autophagic destruction by targeting the DC-SIGN-TLR2 axis for more favorable intracellular survival. Conversely, FimA fimbriae act as Toll-like receptor (TLR) 2 agonists to promote DC autophagy. By regulating the expression of both fimbriae, *P. gingivalis* achieves immune evasion. Nevertheless, the abovementioned ideas are primarily based on the results of molecular mechanism studies. Further in vivo experiments are imperative to verify the hematogenous dissemination of *P. gingivalis*.

## 6. Mechanisms of *P. gingivalis* Involving in IBD

Noteworthy progress has been made in the research exploring the role of *P. gingivalis* in the progression of IBD. This section presents an overview of the potential mechanisms of *P. gingivalis*’s involvement in IBD.

### 6.1. Gut Dysbiosis

Accumulating evidence indicates that gut dysbiosis plays a crucial role in the progression of IBD. Compared to healthy populations, patients with IBD typically exhibit reduced diversity in their gut microbiota, accompanied by a lower proportion of *Firmicutes* and an increased proportion of *Bacteroidetes* and *Actinobacteria* [71,72]. In addition, certain metabolites derived from gut microbiota, including short-chain fatty acids (SCFAs) and bile acids (BAs), are also believed to be associated with the pathogenesis of IBD [73]. 

After oral administration of *P. gingivalis*, mice are found to have a significant reduction in the bacterial diversity of their gut microbiota, along with an increased ratio of *Bacteroidetes* to *Firmicutes* [48,74,75]. These changes align with the characteristics observed in patients with IBD. Nakajima et al. [74] found that at the genus level, *Prevotella* was significantly increased in the intestine of mice following *P. gingivalis* administration. This increase in *Prevotella* is believed to drive chronic intestinal inflammation and exacerbate chemically induced colitis [76,77]. 

The intestinal microbiota is integral to the metabolism of SCFAs and Bas, playing an important role in maintaining the intestinal barrier integrity and immune homeostasis within the host. SCFAs are thought to play a potential anti-inflammatory role in intestinal inflammation, thereby maintaining intestinal barrier integrity [78,79]. In the presence of gut microbiota, primary BAs are converted into secondary BAs, including lithocholic acid and deoxycholic acid. These secondary BAs are involved in immune regulation and exhibit anti-inflammatory activities [80,81]. In the intestine, *Ruminocobaceae*, *Lachnospiraceae*, and *Marvinbryantia* promote the production of SCFAs [82,83,84], and *Clostridium* and *Eubacterium* are involved in the conversion of primary BAs to secondary BAs [85]. The relative abundance of the bacterial species mentioned above was reduced in *P. gingivalis*-treated mice, potentially contributing to the development of intestinal and systemic inflammation [48]. 

### 6.2. Impairment of Intestinal Barrier

As a protective umbrella for the intestine, the intestinal epithelial barrier protects against the invasion of various bacteria and the penetration of toxins. The monolayer of intestinal epithelial cells acts as a physical barrier by forming a tight seal with transmembrane protein complexes. These complexes consist of adhesive molecules such as tight junctions, adherent junctions, and desmosomes. Several studies have shown that the administration of *P. gingivalis* may impair the integrity of the intestinal epithelium. For instance, the administration of *P. gingivalis* to C57BL/6N mice resulted in a significant decrease in the expression levels of the tight junction proteins ZO-1 and occludin in the small intestine, while no such changes were observed in the large intestine [74,75]. Similarly, Tsuzuno et al. [17] revealed that in a DSS-induced colitis model, treatment with *P. gingivalis* exacerbated intestinal inflammation in mice. Further investigations demonstrated that *P. gingivalis* disrupts intestinal barrier function by decreasing the levels of ZO-1 in epithelial cells.

In addition, gut dysbiosis leads to the impairment of the intestinal barrier. Normally, symbiotic bacteria form a microbial barrier on the surface of the mucosal epithelium to resist the invasion of pathogenic microorganisms through colonization resistance and immune response modulation. Gut dysbiosis can result in the disruption of the intestinal microbial barrier function, which increases intestinal permeability and facilitates the invasion of conditioned pathogens. This, in turn, contributes to the inflammatory response in the colon [73]. Furthermore, gut dysbiosis leads to a decrease in intestinal antimicrobial peptides, thereby compromising the intestinal barrier function [86].

### 6.3. Release of Inflammatory Mediators

#### 6.3.1. LPS

LPS is an important component of the cell wall in Gram-negative bacteria. Due to its virulence and the ability to cause unwanted host inflammation, LPS is known as an endotoxin [11]. The LPS of *P. gingivalis* acts as a strong pathogenic agent in periodontal tissues. The virulence of LPS is determined by its lipid A component. By responding to the LPS lipid A component of *P. gingivalis*, host cells generate an inflammatory response in the gingival tissue, creating a favorable environment for pathogens and eventually leading to the progression of periodontal disease [87]. Based on different lipid A structures, LPS can act as an agonist of TLR2 or as an antagonist and/or agonist of TLR4 activation [88,89,90,91,92], causing a series of inflammatory responses. 

Previously, it was suggested that *P. gingivalis* LPS may be involved in the development of intestinal inflammation by inducing a semi-Th2-like response. In a study conducted by Jotwani et al. [93], monocyte-derived DCs (MDDCs) were pulsed with LPS of different oral pathogens. Their findings revealed that, compared to *Escherichia coli* (*E.coli*), *P. gingivalis* LPS induced T cells to produce lower levels of Th1 cytokines and higher levels of Th2 cytokines. An in vivo experiment by Pulendran et al. [94] also supported this conclusion. They demonstrated that *P. gingivalis* LPS-induced T cell responses were characterized by significantly higher levels of IL-5, IL-10, and IL-13 but lower levels of IFN-γ. Conversely, recent studies have proposed that *P. gingivalis*-derived LPS may have certain benefits for colitis [47,95]. Seo et al. [95] found that the administration of *P. gingivalis* LPS attenuated the epithelial damage and lymphocyte infiltration caused by DSS treatment; the DAI score was significantly lower in *P. gingivalis*-treated mice, suggesting a possible ameliorative activity of *P. gingivalis* LPS in colitis. Nevertheless, due to limited evidence, whether LPS plays a positive or negative role in IBD remains to be fully elucidated.

#### 6.3.2. Gingipains

Gingipains are a family of cysteine proteases. They can be divided into arginine-specific gingipains (Rgp, including RgpA and RgpB) and lysine-specific gingipains (Kpg) [11,96]. Gingipains are a major virulence factor of *P. gingivalis*, accounting for 85% of the extracellular protein hydrolytic activity of *P. gingivalis* [97]. They cause dysregulated immune responses and inflammation by activating host matrix metalloproteinases, inactivating immunosuppressive agents, degrading immune factors, and cleaving immune cell receptors [98,99,100,101].

*P. gingivalis* secretes gingipains, which evade the intrinsic immune response by selectively inactivating proinflammatory factors produced by DCs and simultaneously sparing those molecules that enhance their survival. Abdi et al. revealed that *P. gingivalis* significantly reduces the level of IL-12 produced by DCs. Additionally, compared to the W50-NIDCR strain of *P. gingivalis*, the W50-ATCC strain, which proved to secrete fewer gingipains, was less potent at reducing IL-12 production [102].

*P. gingivalis*-derived gingipains are also involved in the destruction of the intestinal barrier, including the mucus layer and intestinal epithelial. As part of the intestinal barrier, the mucus layer plays a crucial role in intestinal homeostasis. Reduced mucus secretion due to decreased levels of goblet cells is a hallmark of human IBD, and mice deficient in the major mucin protein MUC2 develop spontaneous colitis [103,104,105]. Post et al. [106] demonstrated that *P. gingivalis* RgpB can cleave MUC2 mucin in specific sites, resulting in mucus polymer lysis. This process disrupts the intestinal barrier [107,108,109] and contributes to the pathogenesis of IBD [110,111]. Moreover, *P. gingivalis*-derived gingipains may be responsible for the impairment of the intestinal epithelial barrier. Tsuzuno et al. [17] reported that, in contrast to wild-type strains, the gingipain-deficient *P. gingivalis* ATCC33277 mutant failed to reduce ZO-1 protein levels, suggesting that gingipains may be involved in the degradation of ZO-1. However, since the two strains are not isogenic and have different fundamental virulence factors [112,113], further experiments are required to confirm the involvement of gingipains in the disruption of the intestinal epithelium.

### 6.4. Disturbance of Immune Response

#### 6.4.1. Induction of Proinflammatory Cytokines

Inflammatory infiltration in the intestine is also an important pathological change caused by oral administration of *P. gingivalis*. Administration of *P. gingivalis* induces elevated mRNA expression levels of proinflammatory cytokines, including TNF-α, IFN-γ, and IL-6, in the intestine [48,74,75]. Similarly, *P. gingivalis* has been demonstrated to upregulate the expression of IL-6, TNF-α, and IL1-β and exacerbate colitis in DSS-induced mice [17,47]. TNF-α and IFN-γ are typical Th1 cytokines. TNF-α, a pleiotropic cytokine, has been found to be associated with IBD [114,115]. Some studies have observed the overexpression of intestinal TNF-α in patients with CD [116,117,118], leading to the utilization of anti-TNF-α antibody therapy to treat patients with IBD [119,120]. In the intestinal mucosal tissues of IBD patients, the expression level of IFN-γ is highly upregulated, attributed to its immunomodulatory or epithelial effects [121,122]. IL-6 is a key immunomodulator. It acts synergistically with TGF-β to induce the expression of the transcription factor ROR-γ t and promote the differentiation of T helper (Th) 17 cells [123,124]. The aberrant proliferation of Th17 cells and their secretion of a large number of specific cytokines play an important role in the pathogenesis of IBD [125,126]. An elevated level of IL-6 has been found in patients with IBD [127,128]. The inhibition of IL-6 activity by blocking receptors or cytokines with monoclonal antibodies has proved effective in animal models of colitis and in small therapeutic trials for CD [129,130]. IL-1β has the ability to induce proinflammatory innate lymphoid cells and Th17 cells [131]. It also acts as a Th1 skewing factor for the generation of Th1/Th17 cells, which accumulate in the intestine of IBD patients [132,133,134,135,136]. Taken together, *P. gingivalis* has a colitogenic potency, capable of inducing the expression of intestinal proinflammatory cytokines. However, the exact mechanism underlying the induction of these proinflammatory factors in the intestine remains elusive and warrants further discussion.

#### 6.4.2. Upregulation of Th17/Treg Ratio

IBD is a non-specific mucosal inflammatory disease. Intrinsically, it is a CD4+ T cell-mediated disturbance in the balance between the microbial ecosystem and the host immune system, with a key role played by the equilibrium between Th17 cells and Treg cells [6]. Both Th17 and Treg cells are differentiated from naive CD4+ cells.

Th17 cells promote tissue inflammation, while Treg cells suppress autoimmunity in IBD; they are functionally related through differentiation and suppression [137]. Th17 cells and Treg cells share a common signaling pathway mediated by TGF-β. In the presence of IL-6 or IL-21, naive CD4+T cells differentiate into Th17 cells, while in the absence of proinflammatory cytokines, naive CD4+T cells differentiate into Treg cells [138]. The disruption of the Th17/Treg balance can lead to the development of various autoimmune diseases, including IBD. 

Jia et al. [139] found that the cell fragments of *P. gingivalis*, obtained by ultrasonic treatment, upregulated the expression of the Th17-associated transcription factor RoRγt and enhanced the production of the proinflammatory cytokines IL-17 and IL-6 through the TLR4 pathway. Conversely, they downregulated the expression of the Treg transcription factor Foxp3 and suppressed the production of anti-inflammatory factors TGF-β and IL-10. Further in vivo experiments showed that *P. gingivalis*-stimulated CD4+ T cells exacerbated DSS-induced colitis by increasing the Th17/Treg ratio in colonic and lamina propria lymphocytes, which was achieved through the JAK-STAT signaling pathway.

However, the detailed mechanisms underlying how *P. gingivalis* affects Th17/Treg homeostasis and ultimately leads to the development of IBD still need to be investigated. Abnormalities in the Th17/Treg differentiation ratio in the context of IBD are attributed to various factors, including the regulation of inflammatory cytokine expression, influence on BAs release, and induction of intestinal flora dysbiosis [137]. Whether *P. gingivalis* is involved in one or more of these mechanisms warrants further investigation.

The identified mechanisms by which Pg is involved in IBD are depicted in Figure 1.

## 7. Perspectives

Periodontitis and IBD, both complex chronic inflammatory diseases characterized by abnormal host immune responses and microbiota dysbiosis, exhibit a high risk of coexistence and are potentially related. Recently, epidemiological, translational medicine and basic science studies have begun to elucidate the oral–gut association behind periodontitis and IBD [35]. 

In this article, we focus on the role of *P. gingivalis* in linking periodontitis and IBD through the oral–gut axis. We summarize the routes of *P. gingivalis* gut translocation and outline the possible mechanisms of its involvement in the pathogenesis of IBD, which contributes to the search for novel biomarkers and potential therapeutic treatments for IBD. A recent study reported that KYT-36, as a highly selective inhibitor peptide against gingipains, may have promising applications in the treatment of IBD and periodontitis [140]. Furthermore, Jia et al. [139] found that *Lactobacillus rhamnosus* GG (LGG) could decrease the IL-17+Th17 ratio and increase the CD25+Foxp3+Treg ratio through the TLR2 pathway, reversing the increased Th17/Treg ratio caused by *P. gingivalis* and alleviating DSS-induced colitis. This provides a theoretical basis for subsequent investigations on the treatment of inflammatory diseases. 

In addition to gut translocation and ectopic colonization of oral pathobionts, oral-derived immune cells can also be involved in the development of intestinal inflammation through the oral–gut axis. During periodontal inflammation, oral draining lymph nodes produce Th17 cells that recognize oral pathobionts. Oral pathobiont-reactive Th17 cells express intestinal homing molecules, such as CCR9 and α4β7. When oral-derived Th17 cells reach the intestine, they can be activated by translocated oral pathobionts and promote the development of colitis [37]. Given that oral pathobionts, such as *P. gingivalis*, upregulate IL1-β expression [17] and are involved in regulating Th17 differentiation [139], it is likely that microbial and immune pathways work synergistically to exacerbate intestinal inflammation.

To better elucidate the correlation between periodontitis and IBD, further clinical evidence is required to assess the importance of the oral–intestinal axis in the development of intestinal inflammation. Additionally, more research efforts are needed to explore the microbiological and immunological associations between periodontitis and IBD, thus providing new targets for the diagnosis and treatment of periodontitis and IBD.

## Figures and Tables

**Figure 1 biomedicines-12-00685-f001:**
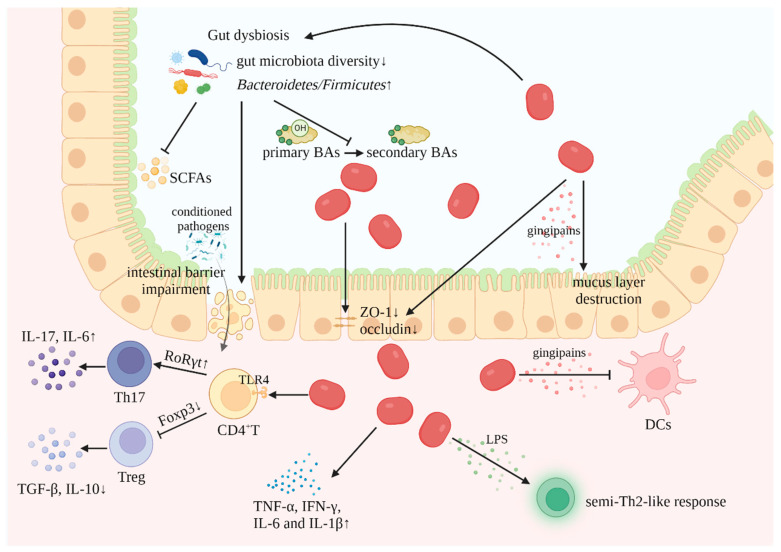
Mechanisms of *P. gingivalis* involvement in IBD (created with BioRender.com). (1) Gut dysbiosis: The reduced diversity of gut microbiota, accompanied by the increased ratio of *Bacteroidetes* to *Firmicutes*, leads to decreased production of SCFAs and inhibits the conversion of BAs. (2) Impairment of intestinal barrier: *P. gingivalis* decreases the expression level of the tight junction proteins ZO-1 and occludin, disrupting the epithelial barrier; meanwhile, gut dysbiosis contributes to the impairment of the intestinal barrier, increasing the invasion of conditioned pathogens. (3) Release of inflammatory mediators: *P. gingivalis*-derived LPS induces a semi-Th2-like response; in addition, *P. gingivalis* secretes gingipains that evade the intrinsic immune response by inhibiting DCs and causing the destruction of the intestinal barrier. (4) Disturbance of immune response: *P. gingivalis* induces the expression of proinflammatory cytokines; in addition, it upregulates the Th17/Treg ratio by regulating the transcription factors RoRγt and Foxp3. (SCFAs, short-chain fatty acids; BAs, bile acids; LPS, lipopolysaccharides; DCs, dendritic cells).

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
