# Peer review of "Linking Periodontitis with Inflammatory Bowel Disease through the Oral–Gut Axis: The Potential Role of Porphyromonas gingivalis"

_biomedicines, 2024, doi:10.3390/biomedicines12030685_

Round 1

Reviewer 1 Report

Comments and Suggestions for Authors

Line 35- " parenteral symptoms"- please clarify?

Porphyromonas gingivalis should be italicized throughout the paper

Line 47- " Thanks to"- please rewrite, does not sound academic

Methodology section regarding literature selection is missing

Line 60- Gram negative not gram negative

3.2 section- microbiome differences in old and young people should be described as epidemiology of IBD differs in older age

Comments on the Quality of English Language

English needs revision in multiple aspects

Author Response

Reviewer 1

Comments 1: Line 35- " parenteral symptoms"- please clarify?

Response: Thanks for the reviewer's comments. We have described parenteral symptoms in more detail and given examples of some diseases in the paragraph. Thank you very much!

Comments 2: Porphyromonas gingivalis should be italicized throughout the paper

Response: Thanks for the reviewer's comments. We have revised the font of the bacterial names to italic throughout the paper. Thank you very much!

Comments 3: Line 47- " Thanks to"- please rewrite, does not sound academic

Response: Thanks for the reviewer's comments. We have revised this expression according to your suggestion. Thank you very much!

Comments 4: Methodology section regarding literature selection is missing

Response: Thanks for the reviewer's comments. Since we do not conduct a systematic review, we did not provide a detailed explanation of the methodology and literature screening process in the article. We also read many other narrative reviews published in Biomedicines and other journals, and most of them did not elaborate on the above content related to methodology. Therefore, we do not present this section in the article. Thank you very much!

Comments 5: Line 60- Gram negative not gram negative

Response: Thanks for the reviewer's comments. We have corrected this error according to your suggestion. Thank you very much!

Comments 6: 3.2 section- microbiome differences in old and young people should be described as epidemiology of IBD differs in older age

Response: Thanks for the reviewer's comments. Following your suggestions, we further investigated the literature on changes in the oral microbiota with age and hypothesized that this phenomenon correlates with the prevalence of IBD at different ages, and we elaborate on the above in 3.2 section. Thank you very much for your useful advice!

Reviewer 2 Report

Comments and Suggestions for Authors

Dear Authors!

Bravo! Your topic choice is highly valued as it is of the highest importance and sadly nevertheless still mostly ignored by the profession. Your research is detailed and well-assembled in your presentation. This project deserves to attract attention and help guide further urgently needed research in diagnosis, treatment guidance, and treatment goal accomplishment. Well done!

Author Response

Comments 1: Bravo! Your topic choice is highly valued as it is of the highest importance and sadly nevertheless still mostly ignored by the profession. Your research is detailed and well-assembled in your presentation. This project deserves to attract attention and help guide further urgently needed research in diagnosis, treatment guidance, and treatment goal accomplishment. Well done!

Response: Thank you very much! We really appreciate the reviewer’s positive comments on our work.

Reviewer 3 Report

Comments and Suggestions for Authors

This review explores P. gingivalis' role, highlighting its impact on gut dysbiosis, barrier function, inflammation, and immune response in IBD pathogenesis.

It is a good and comprehensive work.

Bacterial names should be in italic.

Author Response

Comments 1: This review explores P. gingivalis' role, highlighting its impact on gut dysbiosis, barrier function, inflammation, and immune response in IBD pathogenesis. It is a good and comprehensive work. Bacterial names should be in italic.

Response: We appreciate the reviewer’s positive comments on our work and we have revised the font of the bacterial names to italic throughout the paper. Thank you very much!

Reviewer 4 Report

Comments and Suggestions for Authors

The manuscript prepared by the authors attempts to delve into recent studies on the bidirectional relationship between periodontitis and IBD, and the function of P. gingivalis as a bridge between both diseases. However, the structure of the manuscript is unclear. Authors should better substantiate the objective of their study and make clear whether what they are doing is a systematic review, a scoping review or a narrative review.

The authors must explain what methodology they have followed, how they have selected the articles they refer to and what other articles they have excluded from this "review".

The authors do not refer to studies in which inflammatory bowel disease is related to caries or apical periodontitis. This may represent a limitation of the study. A paragraph should be included in the discussion discussing this possible limitation.

Author Response

Comments 1: The manuscript prepared by the authors attempts to delve into recent studies on the bidirectional relationship between periodontitis and IBD, and the function of P. gingivalis as a bridge between both diseases. However, the structure of the manuscript is unclear. Authors should better substantiate the objective of their study and make clear whether what they are doing is a systematic review, a scoping review or a narrative review.

Response: Thanks for the reviewer's comments. In order to make the logical structure of the article clearer to understand, we have made some revisions in the Abstract section to clarify what we are doing in this narrative review. We also explain the logical structure of the article in the last paragraph of the Introduction section. Thank you very much for your useful suggestions!

Comments 2: The authors must explain what methodology they have followed, how they have selected the articles they refer to and what other articles they have excluded from this "review".

Response: Thanks for the reviewer's comments. Since we do not conduct a systematic review, we do not provide a detailed explanation of the methodology and literature screening process in the article. We also read many other narrative reviews published in Biomedicines and other journals, and most of them did not elaborate on the above content related to methodology. Therefore, we do not present this section in the article. Thank you very much!

Comments 3: The authors do not refer to studies in which inflammatory bowel disease is related to caries or apical periodontitis. This may represent a limitation of the study. A paragraph should be included in the discussion discussing this possible limitation.

Response: Thanks for the reviewer's comments. We have carefully considered your suggestions. However, our article focuses on the interrelationship between periodontitis and IBD and discusses the mechanisms by which P. gingivalis is involved, and therefore, we do not find a discussion of the correlation between IBD and other oral diseases such as caries and apical periodontitis to be necessary in our article, so we do not refer to relevant studies or include this point in our discussion as a limitation. Thank you very much!

Round 2

Reviewer 4 Report

Comments and Suggestions for Authors

Modifications made by the authors have improved the manuscript. It can be accepted in its current form.